# Evaluation of Pecan [*Carya illinoinensis* (Wangenh.) K. Koch] Cultivars for Possible Cultivation for Both Fruit and Truffle Production in the Puglia Region, Southeastern Italy

**Giuseppe Ferrara [1],\*** , **Leonardo Lombardini [2]** , **Andrea Mazzeo [1]** and **Giovanni Luigi Bruno [1],\***

[1] Department of Soil, Plant and Food Sciences, University of Bari Aldo Moro, Via G. Amendola 165/A, 70126 Bari, Italy
[2] Department of Horticulture, University of Georgia, 1111 Plant Sciences Bldg, Athens, GA 30602, USA
\* Correspondence: giuseppe.ferrara@uniba.it (G.F.); giovanniluigi.bruno@uniba.it (G.L.B.);
Tel.: +39-0805442979 (G.F.); +39-0805443085 (G.L.B.)

**Abstract:** Yield and different nut parameters were measured for two growing seasons on mature (28–29 years) trees of 11 pecan cultivars grown in an experimental orchard located in the Puglia Region, Southeastern Italy. 'Shoshoni' and 'Shawnee' pecan seedlings were inoculated with three truffle species (*Tuber borchii*—known as the 'whitish truffle', *T. aestivum*—called the 'summer truffle', and *T. melanosporum*—the common 'Black truffle') and investigated for six months. The level of ectomycorrhizal colonization was assessed 6 and 12 months after inoculation. Results indicated that 'Wichita', 'Shoshoni', and 'Pawnee' performed well in the pedoclimatic conditions of the area with a yield higher than 20 kg/tree and a kernel dry weight of ≅3 g. These preliminary yield results suggested that some pecan cultivars could deserve consideration for cultivation in the Puglia Region, whereas others with low yield and a stronger alternate bearing should not be considered. Plant height, number of leaves, chlorophyll content (expressed as a SPAD unit) and stem diameter partially indicated the increase in ecological fitness in truffles-inoculated plants. Successful mycorrhization indicated 'Shoshoni' and 'Shawnee' as suitable to establish ectomycorrhizal symbiosis with *T. aestivum*, *T. borchii*, and *T. melanosporum* cultivation under Puglia climatic conditions. The results also showed that the applied protocol was adequate to obtain healthy mycorrhized seedlings appropriate for commercialization and plantation for truffles production on pecan in the future.

**Keywords:** pecan; truffle; nut; kernel; mycorrhization; ectomycorrhiza; *Tuber borchii*; *Tuber aestivum*; *Tuber melanosporum*

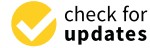



## 1. Introduction

Pecan [*Carya illinoinensis* (Wangenh.) K. Koch] is a deciduous nut tree species in Juglandaceae, native to North America and is the only species of the genus *Carya* with commercial importance [1]. Pecan is cultivated to yield edible fruits (drupaceous nuts) consumed as dried fruit or as processed food, as an ornamental plant, and to promote timber wood production [2,3]. Pecan is a species with a significant alternate bearing showing extreme variability among cultivars [4]. In recent years, demand for pecan kernels has increased worldwide, partly because of the good flavour of the kernel, their high nutritional value or simply an alternative crop to traditional and widely known nuts (pistachios, almonds, walnuts, etc.). Pecans are mainly cultivated in the United States and Mexico, with smaller production in Australia [5], Argentina [6], China [7] and in even fewer amounts in countries such as Israel, Brazil, Peru, and South Africa [8]. In recent years, consumers have been looking for fruits with nutraceutical properties and this led to the increase in consumption of 'minor' fruits such as pomegranate, fig, goji and several berries either fresh or processed [9–12]. Among such fruits, pecans can be mentioned for their beneficial health effects, since they contain phenolic compounds, mono- and polyunsaturated fatty acids,

phytosterols, tocopherols and micronutrients that are associated with a reduced risk of several heart diseases [13]. From a nutritional point of view, pecan nuts are a considerable source of energy (72 g of fat per 100 g of fresh weight), proteins (9.2 g per 100 g of fresh product), omega-6, unsaturated fats that help keep blood cholesterol levels low and contain phenolic compounds with high antioxidant capacity, minerals, and vitamins [13].

In Italy, suitable pedo-climatic conditions for pecan production are in the southern regions such as Puglia and Sicily [3], but pecan could grow well also in the Po Valley (Northern Italy). The Puglia Region, located in Southeastern Italy, is an important horticultural area for the cultivation of the grape and other important fruit crops such as olive, sweet cherry, pomegranate, and almond. The pedoclimatic conditions in the Puglia region are similar to some regions in the USA (i.e., Arizona, New Mexico, Texas) where pecan is cultivated, and some areas in the region could be suitable for pecan cultivation; however, there is very limited scientific information regarding pecan cultivation in this region or even in other regions in Italy.

Symbiosis defines any type of close and long-term biological interaction between two different organisms. The symbiotic organisms live together. This relationship can be beneficial to only one member (commensalism) or both (mutualistic), cause harm to one of the symbiotic organisms (parasitism) or be harmless to both. The symbiotic association between a mycobiont (fungus) and the roots of various plant species is defined as mycorrhiza. Arbuscular mycorrhiza (AM), ectomycorrhiza (ECM), orchid mycorrhiza (OM) and ericoid mycorrhiza (ERM) are differentiated on their structure and function. AM, e.g., those associated with *Rhizophagus intraradices* (N.C. Schenck & G.S. Sm.) C. Walker & A. Schüßler, form fungal hyphae that run parallel to the endodermis inside the root cortex. Fungi forming OM typically produce inside the cells of the orchid primary cortex a peloton, a coil of hyphal loops. ERMs are characterized by fungal coils in the epidermal cells of the fine hair roots of ericaceous species. Fungi evolved in ECMs surround primary and secondary roots as a mantle and form the Hartig net [14,15]. In mycorrhizal symbiosis, fungus and plant balance their mutual needs with huge advantages for the two bionts. For the plant, the hyphae of the fungus allow it to explore a volume of soil outside the roots, improving the absorption of water and mineral elements [15,16]. The plant, in turn, gives the fungus simple sugars and different organic compounds. The mycorrhizal symbiosis influences plant productivity and plant diversity and plays a key role in the cycling of carbon, nitrogen, and phosphorus in ecosystems [15].

Fungi belonging to the genus *Tuber* (Ascomycota, Pezizales) form ECMs associations with a wide range of angiosperms (formerly called Magnoliophyta) and gymnosperms tree species and produce underground ascomata known as truffles [14]. Truffles include species useful in both fresh and processed food, with high gastronomic and economic value for their organoleptic characteristics. In the world, at least 180 truffle species are associated with the genus *Tuber*, 23 of which are present in Italy. *Tuber magnatum* Picco, *T. melanosporum* Vittad., *T. brumale* Vittad. under the two described varieties *moschatum* (Bull.) I.R. Hall, P.K. Buchanan, Y. Wang & Cole and *brumale* Vittad., *T. aestivum* Vittad., *T. borchii* Vittad. (=*T. albidum* Picco), *T. macrosporum* Vittad., *T. mesentericum* Vittad. and *T. uncinatum* Chatin are the truffle species considered edible, harvestable, and marketable according to current Italian regulations. *T. borchii* (known as the whitish truffle), *T. melanosporum* (the common black truffle, Périgord truffle or French black truffle) and *T. aestivum* (called the summer truffle or scorzone truffle) have highly economical and culinary appreciation thanks to their unique flavour and aroma.

*T. aestivum* produces ascocarps from 2 to 10 cm in diameter with pyramidal warts about 3 to 9 mm wide on the peridium, resembling rough bark [17]. *T. borchii* forms ascomata 1–3 cm in size and rounded, with a tuber-like appearance, often bony or irregular [18]. *T. melanosporum* emits round, dark brown ascocarps covered with large spikes. They have a strong, aromatic smell and normally reach a size of up to 10 cm [19].

The truffles of *Tuber* genus are the most interesting forest products from an ecological, hydrogeological, and economic point of view. Specific growing habitat, unpredictable

growth patterns and growing seasons, unique harvesting methods, limited natural resources, and limited shelf life made truffles one of the most expensive foods in the world.

*T. aestivum* and *T. borchii* are widespread in Puglia associated with *Pinus pinea* L., *Pinus halepensis* Mill., *Quercus ilex* L., *Q. pubescens* Willd., *Cistus* spp. in Mediterranean mixed forests, while *T. melanosporum* forms ectomycorrhizal associations with *Q. ilex*, *Q. pubescens*, *Q. cerris* L., *Ostrya carpinifolia* Scop., and *Corylus avellana* L. in cultivated truffle farms. In the case of cultivated truffle farms, an ad hoc environment suitable for truffles is created and supported. Cultural practices involve removing vegetation competing with host plants, ploughing, stone-lining, and irrigation to support plants and truffle growth and development. *Tuber lyonii* Butters, also known as the American brown truffle or the pecan truffle, is the first truffle species described in association with pecan trees in North America orchards [20,21]. *T. melanosporum* and *T. brumale* were also assessed as suitable to establish ectomycorrhizal symbiosis with pecan seedlings [3,22].

Because of the great demand for truffles in the market and the shortage of wild resources, semi-artificial simulation cultivation is a possible strategy for truffle production. In this context, of particular importance is the availability of plant and truffle species well adapted to the cropping area.

The aim of this research was to evaluate quantitative and qualitative characteristics of fruits selected from established pecan cultivars to estimate a possible cultivation in Puglia and in regions with similar pedo-climatic conditions. Moreover, preliminary studies were performed to assess the potential of pecan as suitable species to establish ectomycorrhizal symbiosis with *T. borchii*, *T. aestivum* and *T. melanosporum* grown in Puglia. The generated results could contribute to developing efficient methods for the cultivation of *T. borchii*, *T. aestivum* and *T. melanosporum* on pecan in the future.

## 2. Materials and Methods

### 2.1. Site and Cultivar Evaluation

Puglia has an extension of 19.330 km$^2$, flat for 50% of the territory and around 60% of the region is used for agricultural purposes, mostly for the production of olives (table and oil), sweet cherries, vegetables, and wine and table grapes and also other fruit crops (almond, peach, kiwifruit, pomegranate, fig, etc.) to a less extent. The Region has a typical Mediterranean climate with an average temperature of 15–16 °C, characterized by warm summers (average temperature 25–30 °C) and mild winters (average temperature 6–10 °C), with low rainfall (450–650 mm) spread over all seasons and mostly in autumn-winter, but with heavy rain often occurring in summer in recent years (cloudbursts).

Measurements were collected in 2018 and 2019 on 11 of the 15 cultivars present in the only Italian pecan repository located at the 'P. Martucci' Agricultural Experimental Farm—University of Bari Aldo Moro, Department of Soil, Plant and Food Sciences (Di.S.S.P.A.), Fruit Tree Unit in Valenzano (Bari, Italy), 41°1′35.155″ N, 16°54′6.624″ E (Figure S1). The area has an elevation of 85 m, with average temperature of 15.8 °C and rainfall of 639.2 mm in 2018 and average temperature of 15.5 °C and rainfall of 827.2 mm in 2019.

The repository was established in 1990–1991 using cultivars from the USA. The cultivars used in the study were Pawnee, Wichita, Stuart, Shoshoni, Shawnee, Choctaw, Cheyenne, Green River, Kiowa, Mohawk, and Peruque.

Pecan fruits were harvested from these mature (28–29 years) trees by using poles between 25 September and 20 October. After harvest, the shucks were drop off by hand and the nuts were dried to ≈5% moisture content. The following parameters were measured: (1) Yield/tree (as in shell nuts); (2) whole fruit weight (fresh and dry) and size; (3) nut weight (fresh and dry) and size; (4) kernel weight; (5) hull weight; (6) kernel percentage. Measurement of fresh weight (FW) of all the different parts of the fruits was done and successively samples were dried in a ventilated oven (Mod. BC, ORMA, Milan, Italy) at 65 °C until a constant weight was achieved (DW).

### 2.2. *Mycorrhization Tests*

2.2.1. Pecan Seedlings and Truffle

Shoshoni and Shawnee were selected as pecan cultivars suitable for cultivation to the pedoclimatic conditions of the area. The two cultivars had different yields in the previous seasons, with Shoshoni having high yield while Shawnee having lower yield. Seeds were collected in 2021 from Shoshoni and Shawnee 31-year-old trees present at the 'P. Martucci' Agricultural Experimental Farm. After harvest, seeds were soaked in tap water for 10 days at room temperature and then stratified at 4 °C for 30 days [22]. After stratification, seeds were sown in plastic pots (10 cm width) containing 200 mL of universal soil (Ital-agro: 60% blond peat and 40% vegetable compost soil conditioner; pH 6.5) and placed in a growth chamber at 16 ± 2 °C, with a 16-h photoperiod and 80% air humidity.

Mature ascomata of *T. borchii*, *T. aestivum* collected from natural truffle grounds in the province of Lecce (Italy), and *T. melanosporum* obtained from cultivated truffle farms in the province of Bari (Italy), were used. Truffles were identified by macro- and microscopic morphological analysis. Ascomata were washed under cold tap water and stored in plastic bags at −20 °C. Frozen truffles (about 50 g) surface-sterilized by immersion (3 min) in 70% ethanol to eliminate any contaminants and washed three times (each 3 min) with sterile distilled water, were blended in 100 mL of sterile distilled water with a laboratory blender for 3 min. The suspension obtained was diluted with sterile distilled water to a final concentration of $10^6$ ascospores mL$^{-1}$.

The root systems of the 60–70-days-old pecan seedlings, freed from the cultivation soil, were immersed for about two hours in the spore-slurry of the specified truffle and transplanted into 2 L of a sterile mixture of soil: peat (3:1; *v/v*, pH 8.2 and a good active limestone content). A total of 15 pecan seedlings were inoculated for each treatment (*T. borchii*, *T. aestivum*, and *T. melanosporum*). Additional 15 seedlings were treated with sterile distilled water and used as control. All seedlings were maintained in a greenhouse arranged in a simple randomized block design. Each seedling was watered with about 200 mL every 3 days to keep the appropriate moisture in the substrate. The steps of mycorrhization are illustrated in Figure 1.

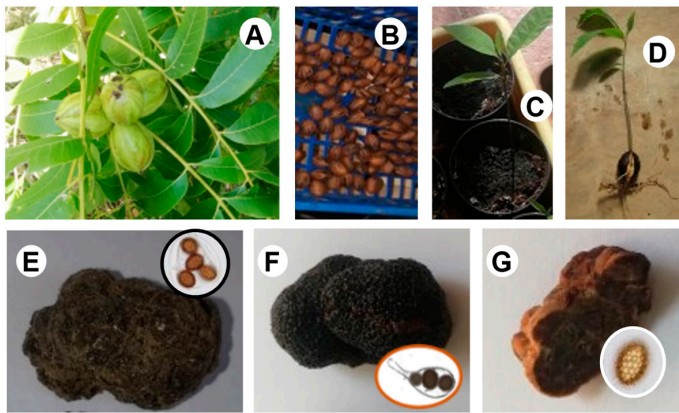

**Figure 1.** The material used in mycorrhization: pecan 'Shoshoni' drupes (**A**), seeds (**B**), seedlings in the emergency phase (**C**) and 'Shawnee' bare root seedling (**D**); ascomata, asci and ascospores of *T. aestivum* (**E**), *T. melanosporum* (**F**), *T. borchii* (**G**).

2.2.2. Eco-Physiology of Pecan-Truffle Interaction

Plant height, stem diameter, number of leaves and chlorophyll content were recorded on the day of transplantation (11 May 2021), 37, and 148 days after transplantation (DAT).

Plant height was assessed with a ruler by measuring the distance between the node of the cotyledon leaves and the apical bud. The stem diameter was measured in the middle portion of the first internode, using a digital vernier caliper. The number of leaves per

plant was also counted. Chlorophyll content was measured with a SPAD 502 instrument (Konica Minolta, Tokyo, Japan).

FW was assessed six months after inoculation using three plants for each treatment. The epigeal part and the root systems of the plants were weighed with an analytical balance to obtain the FW and kept in a ventilated oven at 70 °C up to constant weight for the determination of the DW.

### 2.2.3. Ectomycorrhizal Colonization Levels

The procedure described by Avis et al. (2003) [23] was followed: roots were first gently washed in cold tap water and placed in a glass Petri dish containing distilled water. To estimate the ectomycorrhizal colonization the official method for certifying truffle seedlings produced by commercial nurseries in some Italian regions described by Donnini et al. (2014) [24] was used. The level of ectomycorrhizal colonization was assessed in five seedlings of each truffle species 6 and 12 months after inoculation.

### 2.3. Statistical Analysis

After verifying the assumption of normality (Shapiro-Wilk test: $Pr < W < 0.0001$), data were subjected to analysis of variance (ANOVA) using XLSTAT-Pro software (Addinsoft, Paris, France). The mean and standard deviation (SD) were calculated for each analysed characteristic. The means were separated based on Ryan-Einot-Gabriel-Welsch and Quiot (REGWQ) tests, considering a 95% confidence interval ($p \leq 0.05$).

## 3. Results

### 3.1. Evaluation of Pecan Cultivars

Shuck split is used as a measure of the harvest period of a cultivar. All cultivars reached a 50% shuck split within the middle of October, with Pawnee being early with 50% shuck split by the end-middle of September.

Wichita resulted the cultivar with the highest FW of the fruit (Table 1), either with the hull (≈26 g) or only in-shell (≈9 g), whereas Cheyenne had the lightest fresh fruit weight (≈15 g). Heavy fruits were also noticed for Pawnee and Green River, with more than 24 g.

The FW of the hulls resulted significantly different among the cultivars, with Wichita and Pawnee having the heaviest hulls with a weight of ≈17 g, whereas Cheyenne had the lightest hulls with less than 7 g. Both fruit and hull weight showed significant differences between the two years, with the highest values recorded in 2019. In-shell fruits showed less differences among the cultivars, with Mohawk (≈10 g) having the heaviest in-shell fruit and Peruque with only ≈5 g; however, the values between the two years resulted non-significant (Table 1). Cheyenne was the cultivar with the highest shell FW (4.36 g) and Peruque and Kiowa with the lightest shells (less than 2 g). Only Mohawk and Choctaw had kernel FW greater than 5 g, whereas the kernel FW was in the range of 4.12–4.84 g for the other examined cultivars (Table 1); only Green River and Peruque had kernel FW lower than 4 g. The yield (in-shell nuts) was more than 20 kg/tree in Shoshoni, Wichita and Pawnee; while it was less than 8 kg in Mohawk, Green River, Peruque, Kiowa and Cheyenne, which also showed a strong alternate bearing in the pedoclimatic conditions of the experimental site (Table 1).

Fruits DW (Table 2) followed a similar trend to FW with some remarkable differences. Pawnee was the cultivar with the heaviest hull (≈3 g), whereas Peruque, Shoshoni, Kiowa and Cheyenne had a hull with a weight less than 2 g. Mohawk had the heaviest nuts (>7 g), but all the cultivars had nuts with a weight greater than 5 g, with Stuart, Shawnee and Cheyenne producing in-shell nuts of more than 6 g (Table 2). Stuart was also the cultivar with the heaviest shell (3.52 g); Kiowa and Peruque had a shell weight less than 2 g. Shawnee and Mohawk produced kernels with the highest dry weight, but other cultivars had a kernel with a weight greater than 3 g (Table 2). The kernel percentage was around 55–58% for many cultivars and only Cheyenne, Choctaw and Stuart had values lower than 50% (Table 2).

**Table 1.** Fresh weight of fruit parameters and yield of in-shell nuts as the average of 2018 and 2019 growing seasons [a].

| Cultivar | Fruit (g) | Hull (g) | In-Shell (g) | Shell (g) | Kernel (g) | Yield/Tree (kg) |
|---|---|---|---|---|---|---|
| Cheyenne | 14.89 d | 6.61 c | 8.46 ab | 4.36 a | 4.39 bd | 4.7 d |
| Choctaw | 21.21 ad | 13.43 ab | 7.92 ab | 3.35 bc | 5.29 ab | 16.7 b |
| Green River | 24.68 ab | 16.52 a | 7.98 ab | 3.13 cd | 3.95 de | 5.9 d |
| Kiowa | 16.51 cd | 9.68 bc | 6.68 bc | 1.97 e | 4.12 ce | 5.1 d |
| Mohawk | 19.87 bd | 9.65 bc | 10.23 a | 4.27 a | 5.91 a | 7.1 d |
| Pawnee | 26.45 a | 16.80 a | 8.79 ab | 3.22 bd | 4.84 b | 21.8 ab |
| Peruque | 20.01 ad | 14.31 ab | 5.57 c | 1.84 e | 3.23 e | 5.5 d |
| Shawnee | 21.71 ac | 10.74 bc | 8.77 ab | 2.91 cd | 4.75 bc | 12.3 c |
| Shoshoni | 17.71 cd | 9.28 bc | 8.14 ab | 2.78 d | 4.24 bd | 24.6 a |
| Stuart | 22.86 ac | 13.56 ab | 9.26 ab | 3.59 b | 4.67 bc | 19.7 ab |
| Wichita | 26.61 a | 17.75 a | 8.77 ab | 2.97 cd | 4.52 bd | 23.2 a |
| YEAR | 0.002 | <0.0001 | 0.123 | 0.311 | 0.00019 | <0.0001 |
| CULTIVAR | <0.0001 | <0.0001 | 0.018 | <0.0001 | 0.005 | <0.0001 |
| YEAR × CULTIVAR | <0.0001 | <0.0001 | <0.0001 | <0.0001 | 0.00024 | <0.0001 |

[a] Within each column means with different letters are significantly different according to the REGWQ test at $p < 0.05$.

**Table 2.** Dry weight of fruit parameters as the average of 2018 and 2019 growing seasons [a].

| Cultivar | Hull (g) | In-Shell (g) | Shell (g) | Kernel (g) | Kernel (%) |
|---|---|---|---|---|---|
| Cheyenne | 1.72 cd | 6.03 ab | 3.01 ac | 2.91 ab | 48.3 b |
| Choctaw | 2.38 ad | 5.36 bc | 2.84 ac | 2.46 b | 45.9 b |
| Green River | 3.02 a | 5.83 ac | 2.62 bd | 3.36 ab | 57.6 a |
| Kiowa | 1.58 d | 5.30 bc | 1.90 de | 3.09 ab | 58.3 a |
| Mohawk | 2.68 ac | 7.33 a | 3.31 ab | 3.97 a | 54.2 a |
| Pawnee | 3.09 a | 5.28 bc | 2.40 ce | 2.95 ab | 55.9 a |
| Peruque | 1.88 bd | 3.92 c | 1.64 e | 2.27 b | 57.9 a |
| Shawnee | 2.26 ad | 6.13 ab | 2.63 bd | 3.52 a | 57.4 a |
| Shoshoni | 1.71 cd | 5.60 bc | 2.52 bd | 3.12 ab | 55.7 a |
| Stuart | 2.64 ac | 6.44 ab | 3.52 a | 2.94 ab | 45.7 b |
| Wichita | 2.73 ab | 5.43 bc | 2.38 ce | 3.07 ab | 56.5 a |
| YEAR | 0.164 | 0.001 | <0.0001 | 0.021 | 0.029 |
| CULTIVAR | <0.0001 | 0.001 | <0.0001 | 0.0002 | 0.001 |
| YEAR × CULTIVAR | <0.0001 | <0.0001 | <0.0001 | <0.0001 | <0.0001 |

[a] Within each column means with different letters are significantly different according to the REGWQ test at $p < 0.05$.

The whole fruit length ranged from 36.7 mm in Shoshoni up to 57.5 mm in Wichita; whereas the width had a narrower range, from 26.7 mm in Shawnee up to 34.3 in Green River (Table 3). The length of in-shell nuts ranged from 44.6 mm (Wichita) down to ≈29 mm (Peruque and Shoshoni), whereas the width was in the range of 20–23 mm, with Mohawk having the widest nuts (24.5 mm) (Table 3).

**Table 3.** Size of fruit parameters as the average of 2018 and 2019 growing seasons [a].

| Cultivar | Fruit Length (mm) | Fruit Width (mm) | In-Shell Length (mm) | Shelled Width (mm) |
|---|---|---|---|---|
| Cheyenne | 38.56 ef | 28.17 e | 31.50 e | 23.55 b |
| Choctaw | 46.56 c | 31.07 bc | 37.68 c | 22.79 c |
| Green River | 45.88 c | 34.33 a | 34.80 d | 23.18 bc |
| Kiowa | 43.93 d | 28.34 e | 35.43 d | 19.65 e |
| Mohawk | 47.22 c | 30.16 cd | 40.81 b | 24.47 a |
| Pawnee | 52.59 b | 33.48 a | 40.44 b | 22.76 c |
| Peruque | 40.39 e | 33.49 a | 28.92 f | 20.34 d |
| Shawnee | 52.84 b | 26.66 f | 44.17 a | 20.47 d |
| Shoshoni | 36.67 f | 29.57 d | 29.43 f | 23.18 bc |
| Stuart | 44.15 d | 32.09 b | 35.37 d | 22.77 c |
| Wichita | 57.49 a | 31.40 b | 44.60 a | 20.77 d |
| YEAR | <0.0001 | <0.0001 | <0.0001 | <0.0001 |
| CULTIVAR | <0.0001 | <0.0001 | <0.0001 | <0.0001 |
| YEAR × CULTIVAR | <0.0001 | <0.0001 | <0.0001 | <0.0001 |

[a] Within each column means with different letters are significantly different according to the REGWQ test at $p < 0.05$.

### 3.2. Mycorrhization Tests

3.2.1. Ectomycorrhizal Colonization Levels

Shoshoni and Shawnee colonization with *T. borchii*, *T. melanosporum* and *T. aestivum* was successful. Seedlings appeared healthy and well-structured. The selected plants (Figure 2) were healthy and developed both good shoots and roots.

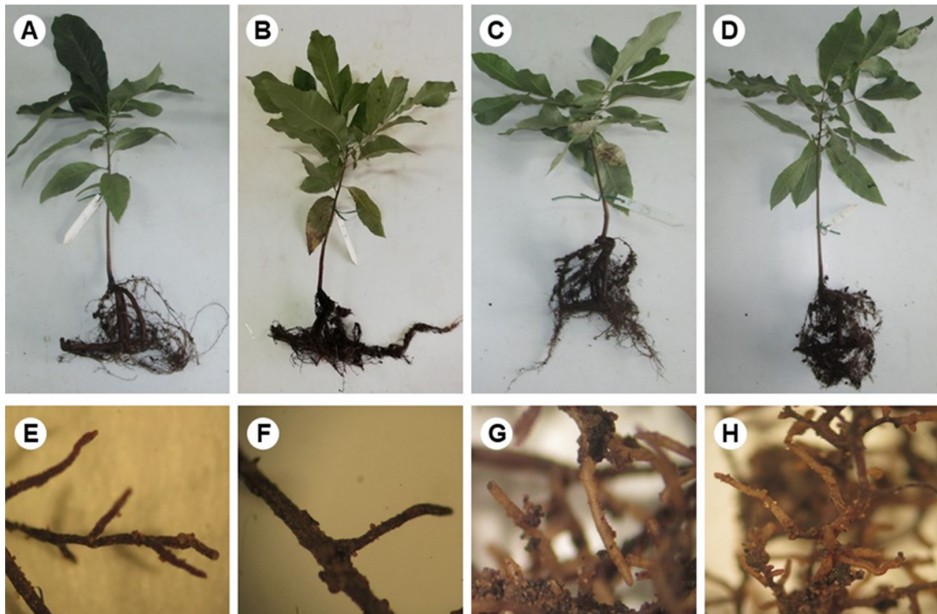

**Figure 2.** 'Shoshoni' seedlings (**A–D**) and roots (**E–H**) after six months of treatments with water (**A,E**), *T. borchii* (**B,F**), *T. aestivum* (**C,G**), or *T. melanosporum* (**D,H**). Roots magnification (**E–H**) 80×.

At the stereomicroscope, the seedlings of the control plants (Figure 2E) and those inoculated with *T. borchii* (Figure 2F) showed little development of roots. The roots of the plant seedlings inoculated with *T. aestivum* (Figure 2G) and *T. melanosporum* (Figure 2H) had an abundant presence of roots on which the formation of the mycoclena could be glimpsed.

The percentage of pecan mycorrhization, assessed 12 months after the inoculation, ranged from 41 to 64% for all the observed seedlings and truffle species inoculated (Table 4).

**Table 4.** Percentage of pecan mycorrhization assessed 12 months after the inoculation with *Tuber borchii*, *T. melanosporum*, *T. aestivum* or control (water) [a].

| Seedling | *T. borchii* | *T. melanosporum* | *T. aestivum* | Control (Water) |
|---|---|---|---|---|
| Shoshoni | 41–(52 ± 9.7)–64 | 45–(52.6 ± 7.1)–60 | 42–(51 ± 8.4)–60 | 0 |
| Shawnee | 45–(54.2 ± 7.1)–60 | 42–(54 ± 7.1)–60 | 42–(53.8 ± 5.4)–62 | 0 |

[a] Values represents the minimum, (the average ± standard deviation) and the maximum referred to 5 seedlings.

3.2.2. Growth and Physiology of Pecan Seedlings

As expected, the height of plant seedlings increased with time (Figure 3A). After 148 days, the control plants and those inoculated with *T. borchii* were 150% taller, while those treated with *T. melanosporum* and *T. aestivum* were 200% and 170% taller, respectively. Stem diameter (Figure 3B) doubled during the first 148 days. No significant differences were recorded between the two tested cultivars and among the four treatments except for *T. borchii* after 148 days.

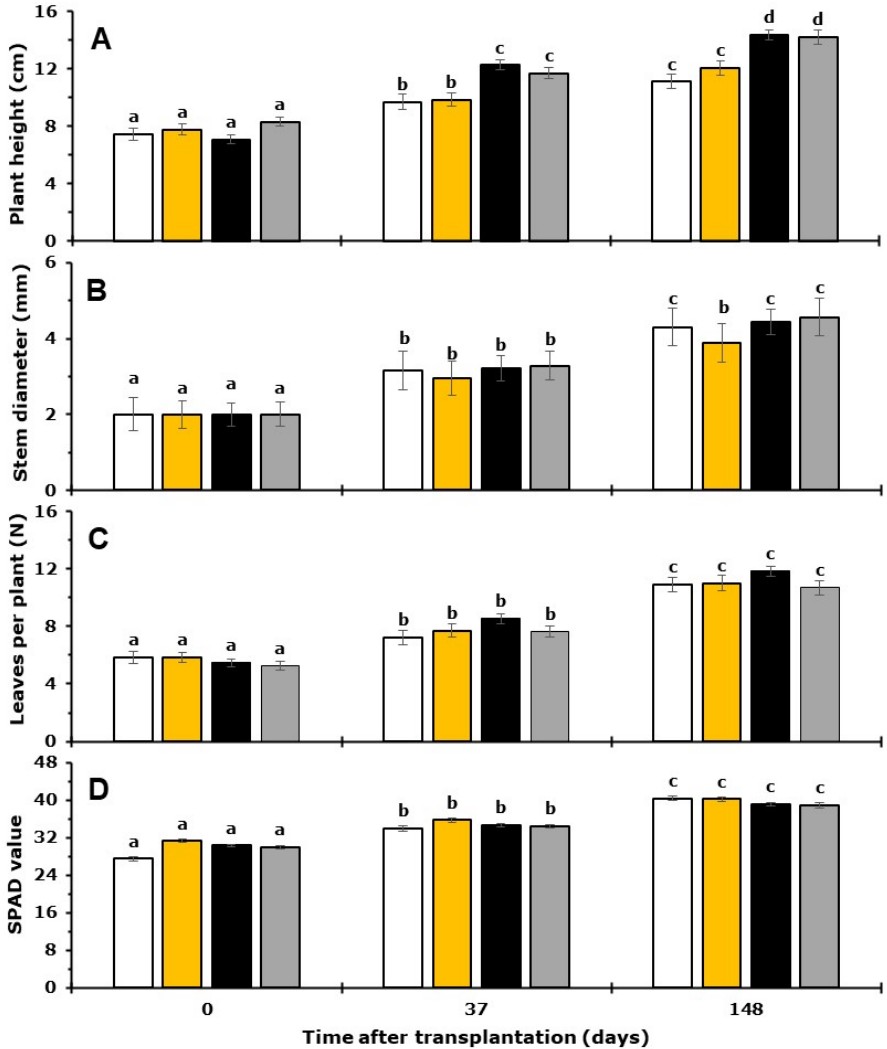

**Figure 3.** Height (**A**), stem diameter (**B**), number of leaves (**C**) and chlorophyll content (**D**) of Shoshoni and Shawnee seedlings inoculated with *Tuber borchii* (🟨), *T. melanosporum* (⬛) or *T. aestivum* (⬜) and used as a control (⬜) measured at the time of transplantation (0), after 37 and 148 days from transplantation. Each histogram represents the average of 13 plants ± standard deviation. Values with the same letter are not significantly different for the REGWQ test at $p < 0.05$.

The number of leaves per plant (Figure 3C) and SPAD index (Figure 3D) increased with time but there were no differences among treatments. No significant differences were recorded between the two tested cultivars.

The greatest FW (from 74.4 to 79.7 g for plants inoculated with *T. borchii* and *T. melanosporum*, respectively) was recorded for the root systems of plants treated with the spore suspension of the three truffle species (Figure 4), while the smallest (about 34.9 g) was in the control plants. A similar pattern was recorded for the shoots (Figure 4). The control plants (13 g) had the lowest shoot dry mass. The greatest dry mass (25.5 g) was measured in plants inoculated with *T. aestivum* (Figure 4). No significant differences were recorded between the two tested cultivars.

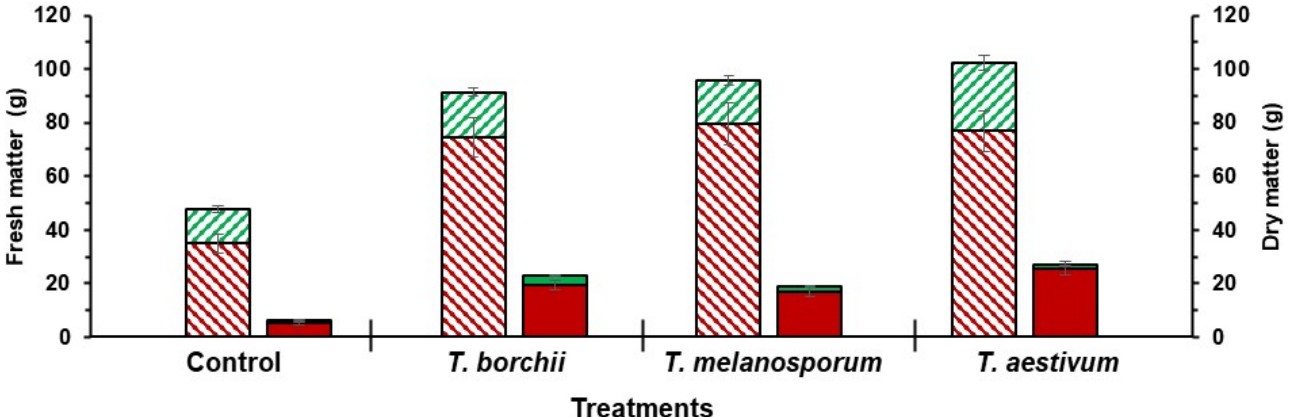

**Figure 4.** Fresh (striped) and dry (solid) matter of root (■) and shoot (■) systems of Shoshoni and Shawnee seedlings after six months of treatments with water (Control), *Tuber borchii, T. melanosporum* or *T. aestivum*. Each value is the average of 3 replications ± SD.

## 4. Discussion

### 4.1. Evaluation of Pecan Cultivars

The yield of the tested cultivars resulted generally lower with respect to the value reported in the USA for similar and different pecan cultivars [4,25]. Probably this difference was a consequence of the management practices used in the two countries and the pedo-climatic conditions of the different sites, in particular a higher rain amount in Southern Georgia with respect to Puglia (400–500 mm higher). However, some cultivars, such as Wichita, Pawnee, Shoshoni, and Choctaw, showed good yield results in the conditions of the Puglia region. Wichita and Pawnee had the longer nuts, and large sized nuts have always been valued more highly by consumers [26]. Wichita performed well also in another Mediterranean country, Egypt, with satisfactory nut yields per tree as in our trials (16.5 kg/tree in Egypt vs. 23.3 kg/tree in Puglia), although our trees were older, but the nut parameters (weight and size) of Wichita grown in Puglia resulted lower than the values reported under drip irrigation in Egypt [27]. In this latter area, drip irrigation reached a volume of 12,000 m³/ha which could partly explain the more favourable nut parameters measured in Egypt with respect to Puglia [27]. A recent study in Anatolia (Turkey) on five pecan cultivars, including Mohawk, reported nut width and length very similar to our values, whereas the kernel weight was slightly higher in Turkey [28]. However, for the trial conducted in the Anatolian pecan repository, no information about either pedo-climatic conditions or cultural practices adopted are reported (i.e., volume of irrigation) [28].

The kernel percentage of Stuart (45.7%) resulted almost identical to what reported in a 12-year evaluation field in Georgia, i.e., 45.3% [4], as indicating a very stable feature of the cultivar. Percent kernel is a clear indicator of the quality of the nuts and is a function of kernel development during the season as well as shell thickness and nut size [4,29]. Our values of kernel percentage were higher than 50% for 8 out of the 11 tested cultivars, and similar high kernel percentages were reported in Georgia on nine pecan cultivars [4]. Ker-

nel percentage can be reduced in some years by scab or other pest infections which stop the kernel development during the season [4]. A recent investigation carried out in Uruguay on the nutritional composition of several pecan cultivars reported values of kernel percentage very similar for pecan cultivars tested in both Uruguay and Italy, confirming the conservative aspect of the feature [30].

The nut weight was less than 8.3 g for some cultivars since this is considered the minimum weight to obtain higher prices on the market [31] and such cultivars (i.e., Peruque, Kiowa) should not be considered for cultivation in the area and others should be preferred (Pawnee, Stuart, Wichita, etc.). Similar nut weight and size together with the kernel weight were measured in Turkey (Antalya) for the cultivars Mohawk and Wichita, whereas Choctaw and Stuart presented lower values in the Puglia region; however, our data are on two seasons and the different pedo-climatic conditions would have partly affected the results because Antalya has an average rainfall of >1000 mm per year [32], much higher than Puglia.

Overall, this two-year trial brings information about cultivars more prone to the pedo-climatic conditions of the Puglia region, with satisfactory yield even with limited water availability during the growing season but with support of irrigation in the most sensitive phenological stages. Alternative waters could be used for irrigation also with the useful support of water sensors as adopted in other species of the Mediterranean basin [33,34]. There are positive prospects for pecan production since the kernels have a pleasant flavour and important nutraceutical properties. Consumers acceptance of pecan quality is generally driven by flavour, interior colour, flavour intensities and even emotional responses more than the strict and simple kernel size, as recently reported in Texas [35]. In this perspective, pecan can be an economic investment that can bring, in medium term, good returns to the farmers as a possible alternative to traditional crops of the area (olive, grape, etc.). A limit for the cultivation of some cultivars can be the alternate bearing and the amount of water needed for irrigation, very different from other species most cultivated in the region (olive, almond, fig), which lead some cultivars to a low yield in 2018. The alternate bearing could be significantly reduced in pecan trees by the application of growth regulators which can also improve the yield, as observed in Mexico for Wichita trees treated with gibberellic acid (GA$_3$), calcium prohexadione (3-oxido-4-propionyl-5-oxo-3cyclohexene-carboxylate) and thidiazuron [1-phenyl-3-(1,2,3 thidiazol-5-yl) urea] [36]. A pecan evaluation of 10 cultivars in Hongzhai region of China, with an average temperature of 16.7 °C and rainfall of 1500 mm, different from Puglia climatic conditions, resulted in heavier nut (10.3 g) and kernel (5.3 g) weights as average [37]. Moreover, pecan plants could be used for the production of timber to be used for furniture, floor, etc.

### 4.2. Mycorrhization

The successful and efficient mycorrhizal synthesis is the basis of the artificial cultivation of truffles. The observations here conducted after six and 12 months after inoculation are promising from the point of view of the success of mycorrhization of pecan seedlings with *T. melanosporum* and especially with *T. aestivum* and *T. borchii*. Stereomicroscope observations showed the actual presence of structures attributable to a process of mycorrhization in the start-up phase.

The results of this study confirm those already obtained by other authors about the possibility of symbiosis between truffles and pecan plants [3,20,22,38].

Pecan trees have a dedicated mutualistic symbiosis with *T. lyonia* in North America orchards [20,21]. Previous studies also demonstrated the mycorrhiza formation on pecan seedlings with *T. melanosporum*, *T. brumale*, *T. borchii* and *T. aestivum* [3,20,22,38]. In the first 12 months of 'Elliott' mycorrhization, *T. melanosporum* and *T. brumale* produced a root colonization level of 37.3 and 34.5%, respectively. After 24 months, the level of mycorrhization became 49.4% and 10.5% for *T. brumale* and *T. melanosporum* respectively, while increase greenhouse contaminants (e.g., *Sphaerosporella brunnea* Alb. & Schwein.) Svrček & Kubička, *Trichophaea woolhopeia* (Cooke & W. Phillips) Quél., *Pulvinula constellatio* (P. Karst.)

Pfister) [3]. Studies found that 0%, 42%, 62% of pecan seedlings after 10 months of inoculation [39], and 50% of pecan seedlings after 6 months of inoculation [20,22] were mycorrhized by *T. macrosporum*, *T. aestivum*, *T. borchii* and *T. lyonii*, respectively.

The colonization level reached by the pecan seedlings here inoculated, ranging from 41 to 64%, is among the highest known mycorrhization levels achieved for the different truffle inoculation procedures. In previous studies, the highest percentage of mycorrhization on pecan was recorded at ≈61% for *T. borchii* and ≈40% for *T. aestivum* [22] or 20–50% of root system for *T. melanosporum* and *T. brumale* [3,22,38–41].

In addition, in our study truffle-inoculated seedlings reached the requirements suitable for seedling commercialization, such as quality of truffle-spore inoculum, quality of the host plant (length of the stem, diameter, well-formed root system), and high level of colonization [24]. The presence of well-colonized truffle-inoculated seedlings is an important feature in agroforestry systems planted for truffle production. Roots well-colonized by truffles will limit the colonization by other ectomycorrhizal fungi in the soil and promote the fruiting of truffles [40–42].

The three truffle species here evaluated showed a higher ability to colonize the roots of pecan thus becoming the most promising symbiotic partner for pecans. In particular *T. melanosporum* supported the growth of the seedlings.

The percentage of mycorrhizal colonization obtained with 1-year mycorrhization cycle also supports the optimal condition offered by the substrate used in our experiment for inoculation and healthy growth of pecan seedlings.

In the tests performed here, importance was given to some eco-physiological parameters useful to discriminate the differences between the state of "health" of the plants and allow physiological discrimination between mycorrhizal and non-mycorrhizal plants.

In the short period of six months, the experimental data collected were not sufficient to allow relevant differentiation between mycorrhizal and non-mycorrhizal plants.

Nevertheless, plant height, number of leaves, chlorophyll content (expressed as a SPAD unit) and stem diameter only partially indicated the increase in growth in inoculated plants. We expect in the long term, once a complete and well-established mycorrhization process will occur, these physiological aspects can be highlighted in the successive years of cultivation.

Successful mycorrhization of two pecan cultivars with three truffle species combinations produced seedlings appropriate for commercialization. 'Shoshoni' and 'Shawnee' pecan appeared to be suitable hosts for *T. aestivum*, *T. borchii*, and *T. melanosporum* and truffle cultivation in the Puglia Region (Southeastern Italy) climatic conditions. The results also showed that the applied protocol was adequate to obtain healthy mycorrhized seedlings suitable for commercialization and plantation as truffle/pecan systems for both truffle and pecan-nuts production and further cultivars could be evaluated.

## 5. Conclusions

Pecan is a promising woody tree to yield delicious, nutritious kernels consumed as a dried fruit or as processed food, and can be used as an ornamental plant, to promote timber wood production. Cultivars such as Wichita, Shoshoni, and Pawnee yielded more than 20 kg of in shell nuts per tree and resulted suitable to the pedo-climatic conditions of the area (Southeastern Italy), characterized by warm summers and an average rainfall of 450–650 mm per year which requires irrigation during the most sensitive stages. Moreover, Shoshoni and Shawnee pecan seedlings were suitable to establish ectomycorrhizal symbiosis with *T. aestivum*, *T. borchii*, and *T. melanosporum* cultivation. The cultivation of pecan mycorrhized plants with commercial truffles could be an important alternative either to recover the rural marginal areas in Mediterranean regions or to improve organic farming production, sustainability, and biodiversity. Further data are needed to confirm and validate these preliminary observations on the truffles in the field.

**Supplementary Materials:** The following supporting information can be downloaded at: https://www.mdpi.com/article/10.3390/horticulturae9020261/s1, Figure S1: The geographical location of the pecan repository in the Puglia region, Southeastern Italy (from Google maps).

**Author Contributions:** Conceptualization, G.F. and G.L.B.; methodology, G.F., G.L.B. and A.M.; formal analysis, G.F. and G.L.B.; data curation, G.F., L.L., A.M. and G.L.B.; writing—original draft preparation, G.F. and G.L.B.; writing—review and editing, G.F., G.L.B. and L.L. All authors have read and agreed to the published version of the manuscript.

**Funding:** This research was partly funded by REGIONE PUGLIA project "Tartufo e tartuficoltura in Puglia", grant number Disspa.Bruno.Tartufo-Rpu—Tartufo e tartuficoltura in Puglia.

**Data Availability Statement:** Not applicable.

**Acknowledgments:** The authors want to thank Nicola Guardavaccaro for the contribution given.

**Conflicts of Interest:** The authors declare no conflict of interest.

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
