# Peer review of "Evaluation of Pecan [Carya illinoinensis (Wangenh.) K. Koch] Cultivars for Possible Cultivation for Both Fruit and Truffle Production in the Puglia Region, Southeastern Italy"

_horticulturae, doi:10.3390/horticulturae9020261_

Round 1

Reviewer 1 Report

Interesting work has been done with valuable results. The structure of the article is well-formed and the different sections are well-described. This research is worth publishing. However, some points need to be done to increase the scientific value of this article.

-          The table related to the analysis of variance should be added.

-          A geographical map of the tested area should be added.

-          Weather characteristics such as temperature, rainfall, etc. of the study area in two cropping seasons should be added.

-          The discussion section needs to be improved. The reasons for the difference in yield and yield components in investigated varieties with other countries should be well discussed.

Author Response

Responses to Reviewer #1

Interesting work has been done with valuable results. The structure of the article is well-formed and the different sections are well-described. This research is worth publishing. However, some points need to be done to increase the scientific value of this article.

We thank the reviewer for these valuable comments and for appreciating the article.

The table related to the analysis of variance should be added.

Such information has been added in the revised ms, at the bottom of each table.

A geographical map of the tested area should be added.

A map of the tested area has been inserted in the revised ms.

Weather characteristics such as temperature, rainfall, etc. of the study area in two cropping seasons should be added.

This information has been now added in the revised ms.

Point 4: The discussion section needs to be improved. The reasons for the difference in yield and yield components in investigated varieties with other countries should be well discussed.

Discussion has been improved in the revised ms.

Reviewer 2 Report

Moderate issues to be solved prior to thee acceptance:

Did you investigate young pecan seedlings or established plants, you mention both, but there is a vast difference. Please clearly state that the seedlings were inoculated to test the inoculation success, but mature trees were investigated for nut characteristics.

Also please explain how did you measure the yielding, there is nothing about it but the final number. 

Lines 250-251: Please show the complete results mentioned here. 

Please amend the conclusion section with actual data. It is too narrative in its current form. Also give some broader recommendations, having in mind the wide audience of the Horticulturae Journal. 

Author Response

Moderate issues to be solved prior to the acceptance:

Did you investigate young pecan seedlings or established plants, you mention both, but there is a vast difference. Please clearly state that the seedlings were inoculated to test the inoculation success, but mature trees were investigated for nut characteristics.

Also please explain how did you measure the yielding, there is nothing about it but the final number.

We investigated adult plants for nut parameters and seedling for truffle inoculation and this is now better reported in the revised ms. The information about yielding has been better explained.

Lines 250-251: Please show the complete results mentioned here.

The results were shown and table 4 was added in the revised ms.

Please amend the conclusion section with actual data. It is too narrative in its current form. Also give some broader recommendations, having in mind the wide audience of the Horticulturae Journal.

The conclusion section has been revised according to the comments of the reviewer.

Reviewer 3 Report

The authors got interesting data, highly likely some cultivars of pecan trees are able to be used in a commercial way. Also, preliminary studies of mycorrhization of pecan seedlings roots by Tuber sp. show that Carya illionolensis might be a suitable host for Truffle fungi. However, I have some comments:

Line 56 - some regions in the USA – what are these regions?

Lines 64-66 – according this information there are two types of mycorrhiza. But it is better to rewrite this sentence with the information from “Mycorrhizal ecology and evolution: the past, the present, and the future” van der Heijen et al. 2015. Not only arbuscular mycorrhiza is intracellular.

Line 91 – Quercus cerris needs to be Q. cerris

The introduction is good but the reason of studying symbiotic relationship of truffles with pecan is not clear. Only lines 87-88 talk about the importance of Tuber genus. In my opinion it would be good to emphasize the economical importance of this symbiotic relationship.

Choose one style of writing cultivar Shoshoni or ‘Shoshoni’

Also, it is not clear why did you decide to study mycorrhiza only with Shoshini and Shawnee? Did you check other cultivars for establishing of mycorrhiza?

The half of the research is devoted to evaluation of pecan trees for possible cultivation in Puglia region in Italy. But nothing is said about some geographical, climate and soil characteristics of this region. Even though it is possible not to pay attention on environmental conditions that are favorable for pecan trees, because there is some information about regions of cultivation in introduction. Puglia region should be somehow characterized, maybe in Materials and Methods. Line 55 says pedoclimatic conditions of Puglia regions are similar to some region in USA. USA is large. How should I know what are these regions, what are the conditions in these regions? How can I understand that pedoclimatic conditions in these regions are similar to Puglia region?

Line 115 – what was the age of trees?

Line 116 – how did to you check the level of moisture?

Line 119 – it seems kernel percentage is the sixth parameter.

Line 129 – what is natural photoperiod for germination? What was the length of day/night, the humidity?

Lines 131, 132 – add a country to Lecce and Bari.

Line 139 – what was the age of pecan seedlings?

Lines 142 -143 – choose something one either ‘whitish truffle’ or T. borchii. There are different names in different parts of the text.

Figures 3 and 4. There should be the same order or names of inoculants. If you use order T. borchii, T. melanosporum, T. aestivum in figure 3, the same should be in figure 4.

Line 268 says that the greatest FW of roots have T. borchii and T. aestivum. Ok. But the pattern is not similar for the shoots. In figure 4 I see that the greatest FW of shoots belongs to T. melanosporum. Then line 273 says that the greatest dry mass has T. aestivum, however it seems that T. melanosporum has the greatest dry mass. Moreover, the result that T. melanosporum has the greatest fresh and dry mass of shoots confirm the results of growth parameters in figure 3.

Lines 357-359 – if T. aestivum and T. borchii show the higher ability to colonize pecan seedling it does not mean that these truffle species are the most promising symbiotic partners.  According to your results pecan seedling grew better with T. melanosporum. And this means that T. melanosporum is the more favorable symbiotic partner. Because even the fungi colonization is low but it is enough to help plant to grow. If there are too much fungi in a plant root, fungi take too much nutrition from plant and begin to behave as a parasite.

Lines 345-349 – these results about root colonization by Truffles should be given in part 3.2.1

Author Response

The authors got interesting data; highly likely some cultivars of pecan trees are able to be used in a commercial way. Also, preliminary studies of mycorrhization of pecan seedlings roots by Tuber sp. show that Carya illionolensis might be a suitable host for Truffle fungi. However, I have some comments:

We thank the reviewer for the positive comments on the article.

Line 56 - some regions in the USA – what are these regions?

The information has been added in the revised ms.

Lines 64-66 – according this information there are two types of mycorrhiza. But it is better to rewrite this sentence with the information from “Mycorrhizal ecology and evolution: the past, the present, and the future” van der Heijen et al. 2015. Not only arbuscular mycorrhiza is intracellular.

The sentence was rewriting following reviewer indication. The reference “van der Heijen et al. 2015. Mycorrhizal ecology and evolution: the past, the present, and the future.” was added and the reference number was corrected.

Line 91 – Quercus cerris needs to be Q. cerris

The name of the genus was abbreviated. The change has been inserted in the revised ms.

The introduction is good but the reason of studying symbiotic relationship of truffles with pecan is not clear. Only lines 87-88 talk about the importance of Tuber genus. In my opinion it would be good to emphasize the economic importance of this symbiotic relationship.

The mycorrhizal symbiosis was emphasized in the introduction.

Choose one style of writing cultivar Shoshoni or ‘Shoshoni’

The style was standardized.

Also, it is not clear why did you decide to study mycorrhiza only with Shoshini and Shawnee? Did you check other cultivars for establishing of mycorrhiza?

Shoshoni and Shawnee were selected as productive (in particular Shoshoni) and suitable to the pedoclimatic conditions of the area. The sentence was inserted on section 2.2.1.

The half of the research is devoted to evaluation of pecan trees for possible cultivation in Puglia region in Italy. But nothing is said about some geographical, climate and soil characteristics of this region. Even though it is possible not to pay attention on environmental conditions that are favorable for pecan trees, because there is some information about regions of cultivation in introduction. Puglia region should be somehow characterized, maybe in Materials and Methods. Line 55 says pedoclimatic conditions of Puglia regions are similar to some region in USA. USA is large. How should I know what are these regions, what are the conditions in these regions? How can I understand that pedoclimatic conditions in these regions are similar to Puglia region?

Climatic information has been now added in the revised text and more information about the characteristic of Puglia are now in the M&M section.

Line 115 – what was the age of trees?

The age of trees was inserted in the revised ms.

Line 116 – how did to you check the level of moisture?

A sample was measured in the oven at the beginning and after the drying process.

Line 119 – it seems kernel percentage is the sixth parameter.

It was revised in the new manuscript.

Line 129 – what is natural photoperiod for germination? What was the length of day/night, the humidity?

Information has been added in the revised ms.

Lines 131, 132 – add a country to Lecce and Bari.

The country to Lecce and Bari was added in the text.

Line 139 – what was the age of pecan seedlings?

The age of pecan seedlings was inserted in the revised ms.

Lines 142 -143 – choose something one either ‘whitish truffle’ or T. borchii. There are different names in different parts of the text.

The name of truffle was standardized. The Latin name was preferred.

Figures 3 and 4. There should be the same order or names of inoculants. If you use order T. borchii, T. melanosporum, T. aestivum in figure 3, the same should be in figure 4.

The order and names of inoculants was uniformed in the figures.

Line 268 says that the greatest FW of roots have T. borchii and T. aestivum. Ok. But the pattern is not similar for the shoots. In figure 4 I see that the greatest FW of shoots belongs to T. melanosporum. Then line 273 says that the greatest dry mass has T. aestivum, however it seems that T. melanosporum has the greatest dry mass. Moreover, the result that T. melanosporum has the greatest fresh and dry mass of shoots confirm the results of growth parameters in figure 3.

The data were controlled, and the periods were rewritten.

Lines 357-359 – if T. aestivum and T. borchii show the higher ability to colonize pecan seedling it does not mean that these truffle species are the most promising symbiotic partners.  According to your results pecan seedling grew better with T. melanosporum. And this means that T. melanosporum is the more favorable symbiotic partner. Because even the fungi colonization is low but it is enough to help plant to grow. If there are too much fungi in a plant root, fungi take too much nutrition from plant and begin to behave as a parasite.

The section was rewritten.

Lines 345-349 – these results about root colonization by Truffles should be given in part 3.2.1

Data on root colonization by Truffles were reported in part 3.2.1. The percentage used represents the levels of mycorrhization on pecan reached by several authors. See references 18,32 and 33.

Round 2

Reviewer 1 Report

According to the corrections that have been made, MS can be accepted.

Author Response

We thank the reviewer for these valuable comments and for accepting the ms for publication.

Reviewer 3 Report

Now the introduction is better. However I’ve read carefully the lines 111-112…are you sure that it is correct?  Now it sounds as T. brumale and T. melanosporum might be hosts for pecan seedlings. They might be hosted by pecan seedlings.

Materials and methods.

Lines 129-130 – what is warm summer and moderate winter? Your readers are the researchers from the whole world, they somehow know what Mediterranean climate is, but they do not know the details. It better and it is not difficult to add details. The temperature in summer is about 27-30°C, and the temperature in winter is about 5-9°C (as some forecast sites says). I see you added the average temperature for 2018 and 2019 years, but it is better also to add winter and summer temperatures in lines 129-130.

Line 141 – As I understand, the age of trees was 28-29 years. It is better to give numbers of the age.

Line 143 – how did you check the level of moisture?

Lines 155-157 – and still it is not clear why did you decide to check mycorrhization on Shoshoni and Shawnee. According to conclusion (line 426) Shoshoni, Wichita and Pawnee are most suitable for producing pecan nuts in Puglia region. What should readers and farmers think?  However if Shawnee (according to you data) is not the best cultivar for Puglia region, maybe the explanation is that you decided to check cultivar with high yield and a cultivar with low yield.

Now I have a question: two cultivars were checked for establishing mycorrhiza, and percentage of mycorrhization is given. But. Are all these data in figures 4 and 5 summarized for both cultivars? What is the cultivar in figure 3?

The order of tuber species is different in figures 3 and 4 and Table 4.

Table 4 is you data. Give an average mean with SD as in figure 4.

Now I do not understand what is going on with figure 5. You changed the order of Tuber species but the order of the data is the same. Now T. borchii has the greatest FW.

Line 161 – what is soil improver green compound?

Lines 384-385 – it would be good to make this part bigger. It is interesting to compare your results and results of other authors. Give some details of these works. And it would be good to mention that also T. lyonii was found on pecan roots by Bonito et al. 2011.

Maybe it is better to replace Figure 1 to supplementary materials.

Author Response

Now the introduction is better. However I’ve read carefully the lines 111-112…are you sure that it is correct? Now it sounds as T. brumale and T. melanosporum might be hosts for pecan seedlings. They might be hosted by pecan seedlings.

We thank the reviewer for these valuable comments. The cited references 3 and 18 report “pecan may represent an attractive alternative host to forest trees for truffle growers given the potential for co-cropping truffles and pecans.” However, the period was better rearranged.

Materials and methods.

Lines 129-130 – what is warm summer and moderate winter? Your readers are the researchers from the whole world, they somehow know what Mediterranean climate is, but they do not know the details. It better and it is not difficult to add details. The temperature in summer is about 27-30°C, and the temperature in winter is about 5-9°C (as some forecast sites says). I see you added the average temperature for 2018 and 2019 years, but it is better also to add winter and summer temperatures in lines 129-130.

As suggested by the reviewer we added more detail on the temperature and rain in lines 129-130 in the revised ms.

Line 141 – As I understand, the age of trees was 28-29 years. It is better to give numbers of the age.

The age of trees was added in the revised ms.

Line 143 – how did you check the level of moisture?

A sample of kernels (after harvest) was weighed at 105 °C to take the initial moisture of the nuts. After the drying process, a successive sample of kernels was weighed for the final moisture.

Lines 155-157 – and still it is not clear why did you decide to check mycorrhization on Shoshoni and Shawnee. According to conclusion (line 426) Shoshoni, Wichita and Pawnee are most suitable for producing pecan nuts in Puglia region. What should readers and farmers think?  However if Shawnee (according to you data) is not the best cultivar for Puglia region, maybe the explanation is that you decided to check cultivar with high yield and a cultivar with low yield.

Now I have a question: two cultivars were checked for establishing mycorrhiza, and percentage of mycorrhization is given. But. Are all these data in figures 4 and 5 summarized for both cultivars? What is the cultivar in figure 3?

The figures have been corrected and the choice of the two cultivars has been now better explained in the revised ms as required by the reviewer. Tha data in figures 4 and 5 are summarized for both cultivars. The cultivar in figure 3 is Shoshoni.

The order of tuber species is different in figures 3 and 4 and Table 4.

The order of Tuber species was modified as suggested.

Table 4 is you data. Give an average mean with SD as in figure 4.

Table 4 was modified accordingly.

Now I do not understand what is going on with figure 5. You changed the order of Tuber species but the order of the data is the same. Now T. borchii has the greatest FW.

We apologize for the inconvenient; the figure has been replaced with the correct one.

Line 161 – what is soil improver green compound?

The “soil improver green compound” has been replaced with “vegetable compost soil conditioner”

Lines 384-385 – it would be good to make this part bigger. It is interesting to compare your results and results of other authors. Give some details of these works. And it would be good to mention that also T. lyonii was found on pecan roots by Bonito et al. 2011.

The comparation was improved and the mention of T. lyonii was also reported. The references 34 (Benucci, et al., 2012. Doi: 10.1007/s00572-011-0413-z) and 35 (Bonito et al., 2012. Doi: 10.1007/s11104-012-1127-5) were also inserted.

Maybe it is better to replace Figure 1 to supplementary materials.

We moved Figure 1 in the supplementary materials as S1 and renumbered the figures in the text.